# Attention for Adversarial Attacks: Learning from your Mistakes

**Florian Jaeckle**[*], **Aleksandr Agadzhanov**[*], **Jingyue Lu, M. Pawan Kumar,**

Department of Engineering Science
University of Oxford
{florian,pawan}@robots.ox.ac.uk, alexagadzhanov97@gmail.com, jingyue.lu@spc.ox.ac.uk

## Abstract

In order to apply Neural Networks in safety-critical settings, such as healthcare or autonomous driving, we need to be able to analyse their robustness against adversarial attacks. As complete verification is often computationally prohibitive, we rely on cheap and effective adversarial attacks to estimate their robustness. However, state-of-the-art adversarial attacks, such as the frequently used PGD attack, often require many random restarts to generate adversarial examples. Each time we perform a restart we ignore all previous unsuccessful runs. In order to alleviate this inefficiency, we propose a method that learns from its mistakes. Specifically, our method uses Graph Neural Networks (GNNs) as an attention mechanism, to greatly reduce the search space for the attacks. The architecture of the GNN is based on the neural network we are attacking, and we perform forward and backward passes though the GNN mimicking the back-propagation algorithm of PGD attacks. The GNN outputs a smaller subspace for the PGD attack to focus on. Using our method, we manage to boost the attacks' performance: the GNN increases the success rate of PGD by over 35% on a recent published dataset used for comparing adversarial attacks, while simultaneously reducing its average computation time.

## Introduction

The success of deep learning relative to traditional machine learning in various areas such as image recognition, natural language processing, or recommendation systems has motivated its usage in more safety-critical applications; examples of which include healthcare and autonomous driving. Despite AI's high level of performance, often beating humans on tasks like computer vision, researchers both in academia and industry have called for machine learning based approaches to be regulated more heavily, especially due their lack of explainability and vulnerability towards malicious attacks. Szegedy et al. (2013) were the first to show that neural networks are susceptible to so-called adversarial attacks. These are methods that slightly perturb an image to get a trained neural network to misclassify it, often with a high level of confidence. Tiny perturbations that are imperceptible to the human eye are often enough to trick the network, as shown in Figure 1. Many different adversarial example

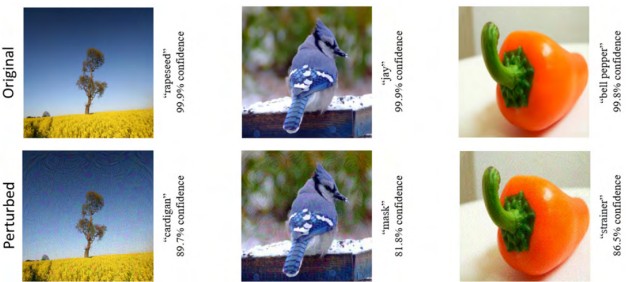

Figure 1: An example of the effect of adversarial attacks on the image classification neural networks taken from Akhtar and Mian (2018). Top: original images which are correctly classified by the neural network with very high levels of confidence. Bottom: same images as above but with small engineered perturbations added to them so that the network now misclassifies them with high levels of confidence.

generation methods have been proposed in the literature. These are important for both evaluating the robustness of neural networks (verification) as well as making them more robust (adversarial training).

In this work we focus on white box attacks; in this setting the attacker has access to the network architecture, as well as its weights. White box attacks are often used for verification and adversarial training as in both cases we have access to the network weights. Moreover, even in a black box setting, where the attacker doesn't have access to the network architecture or weights, we can perform a white box attack on a surrogate model as adversarial examples tend to generalize across different network architectures (Szegedy et al. 2013). Numerous methods aiming to generate adversarial examples have been proposed in the literature. Most of the state-of-the-art approaches are iterative methods using techniques from the standard optimziation literature (Moosavi-Dezfooli, Fawzi, and Frossard 2016; Carlini and Wagner 2017; Dong et al. 2018; Madry et al. 2018). They start at an (often random) initial point and aim to arrive at an adversarial example using many optimization steps. To improve performance most of these attacks use random restarts. We highlight two challenges or weaknesses of this approach. Firstly, the search space is often large as it tends to be high-dimensional, and secondly at every restart we ignore all pre-

vious unsuccessful optimization attempts. We aim to alleviate both of these points using a new attention mechanism. Our method aims to greatly reduce the search space for the attack, by learning from its past mistakes, thereby increasing its chances of finding an adversarial example more quickly.

To this end, we propose to use a Graph Neural Network to reduce the search space for PGD attacks (Madry et al. 2018), one of the most commonly used methods to generate adversarial examples. We treat the Neural Network we are attacking as a graph and mimic the forward-backward nature of the back-propagation algorithm used by PGD using message passing in our GNN. Our GNN takes as input the targeted neural network as well as information from previous unsuccessful PGD attacks and outputs a new input domain that is smaller than the previous search space. Our approach manages to increase the number of properties which we manage to attack successfully by over 35% compared to using PGD with random restarts on a given time frame.

Code for all experiments is available at https://github.com/AleksandrAgadzhanov/GNN_Attention_Mechanism

## Related Work

Many different types of adversarial attacks exist. We focus on white-box image-dependent targeted attacks, as they can be seen as the strongest form of attacks. We work with white-box image-dependent attacks as they are widely used for both verification and adversarial training; we further note that one can create an untargeted attack using an ensemble of targeted ones.

**Adversarial Attack Methods.** Serban, Poll, and Visser (2020) separate adversarial attacks into three main categories. The first, which we focus on in this work, aims to find an adversarial example given an allowed perturbation norm. Examples include the Fast Gradient Sign Method (FGSM) (Goodfellow, Shlens, and Szegedy 2015), Projected Gradient Descent (PGD) (Madry et al. 2018), Iterative FGSM (Kurakin, Goodfellow, and Bengio 2016), and Iterative FGSM with Momentum (Dong et al. 2018). A second type of attacks aims to find an adversarial example with the smallest possible perturbation. The first such attack was proposed by Szegedy et al. (2013) using limited-memory box constrained optimization. Other methods have been proposed by Moosavi-Dezfooli, Fawzi, and Frossard (2016) and Carlini and Wagner (2017). A third line of research focuses on attacks that use machine learning based methods to learn to generate better adversarial examples, such as ATNs (Fischetti and Jo 2018), GAPs (Poursaeed et al. 2018), AdvGANs (Xiao et al. 2018), and AdvGNNs (Jaeckle and Kumar 2021).

**Attention Mechanisms.** Attention mechanism have been widely used in computer vision (Itti, Koch, and Niebur 1998; Ramachandran et al. 2019); their application has been partly inspired by human vision. To the best of our knowledge, very limited work has been done to use attention mechanism for generating adversarial examples. Chen et al. (2017) propose several techniques to reduce the search space and (Cui et al. 2020) use active subspaces to generate adversarial examples. Recently, Wang et al. (2022) published a method

that uses attention information to generate universal adversarial perturbations. Concurrent work by Jia et al. (2021) uses generative networks to improve the initialization for adversarial attacks used for adversarial training. Unlike our method their learnt initialization method is only conditioned on the natural image as well as the gradient information from the target network and doesn't learn from past attacks. Furthermore, the design of their network differs from our Graph Neural Network based approach.

**Graph Neural Networks.** We introduce an attention mechanism utilizing Graph Neural Networks (GNN). GNNs have recently been used in neural network verification in branch-and-bound based algorithms: both to learn a branching strategy (Lu and Kumar 2020) and to learn better bounds (Dvijotham et al. 2018; Gowal et al. 2019; Jaeckle, Lu, and Kumar 2021). Recently Jaeckle and Kumar (2021) used GNNs to learn to generate adversarial examples. We argue that that GNNs are well suited for our problems, as we can treat the neural network we are attacking as a graph and simulate the back-propagation algorithm of adversarial attacks using message passing.

## Problem Definition

We now outline the problem definition along with some standard algorithms to solve it. Throughout this work we write scalars in non-bold italic lowercase or uppercase letters ($\lambda$ or $L$); vectors will be written as bold non-italic lowercase letters ($\mathbf{z}$); the $i$-th element of a vector $\mathbf{z}$ will be denoted as $\mathbf{z}_i$; matrices will be denoted as bold non-italic uppercase letters ($\mathbf{W}$); the element of the matrix $\mathbf{W}$ appearing in the $i$-th row in the $j$-th column will be denoted as $\mathbf{W}_{i,j}$.

We are given an $L$-layer (convolutional) Neural Network $f : \mathbb{R}^d \mapsto \mathbb{R}^m$, that takes as input a $d$-dimensional vector, in our case an image, and outputs an $m$-dimensional vector, corresponding to the $m$ different classes of our classification problem. Given weights $\mathbf{W}^{(j)}$, biases $\mathbf{b}^{(j)}$, and a non-linear activation function $\sigma$, $f$ can be defined as follows:

$$\hat{\mathbf{x}}^{(i+1)} = \mathbf{W}^{(i+1)}\mathbf{x}_i + \mathbf{b}^{i+1}, \quad \text{for } i = 0, \dots, L-1, \quad (1)$$
$$\mathbf{x}_i = \sigma(\hat{\mathbf{x}}_i), \qquad\qquad \text{for } i = 1, \dots, L-1. \quad (2)$$

Throughout this work we use ReLU activations, as they are the most commonly used activation for feed-forward neural networks (Ramachandran, Zoph, and Le 2017). Here $\mathbf{x}^{(0)} \in \mathbb{R}^d$, is the input image, and $\mathbf{x}^{(L)}$ is the output vector. As we are considering image classification problems in this work, $f(\mathbf{x})_j = \mathbf{x}_j^{(L)}$, can be interpreted as the confidence value that the input belongs to the $j$-th class; the image thus gets classified as $\operatorname{argmax}_j f(\mathbf{x})_j$ by $f$.

An adversarial example is an input vector that is close to a natural image but one that gets misclassified by our network. That is, given a real image $\mathbf{x}$ with true label $y$, $\mathbf{x}'$ is an adversarial example if it lies near $\mathbf{x}$ and $f(\mathbf{x})_y < f(\mathbf{x})_{\hat{y}}$ for some incorrect class $\hat{y}$. There are many ways to compute the similarity between two images; we use the infinity norm, as it's been widely used in the literature (Madry et al.

2018; Dong et al. 2018). We require that the distance between $\mathbf{x}$ and $\mathbf{x}'$ is less than some given parameter $\epsilon$, that is $d(\mathbf{x}, \mathbf{x}') := \|\mathbf{x} - \mathbf{x}'\|_\infty := max_j|\mathbf{x}_j - \mathbf{x}'_j| \leq \epsilon$

We can formulate the problem of finding adversarial examples as an optimization problem:

$$max_{\mathbf{x}' \in \mathcal{B}(\mathbf{x}, \epsilon)} \ L(\mathbf{x}', y, \hat{y}) = f(\mathbf{x}')_{\hat{y}} - f(\mathbf{x}')_y, \quad (3)$$

where $\mathcal{B}(\mathbf{x}, \epsilon)$ is an $\epsilon$-sized infinity norm-ball around $\mathbf{x}$:

$$\mathcal{B}(\mathbf{x}, \epsilon) := \{\mathbf{x}' \mid d(\mathbf{x}, \mathbf{x}') \leq \epsilon\}. \quad (4)$$

We call $L$ the adversarial loss and $\mathbf{x}'$ an adversarial example if $L(\mathbf{x}', y, \hat{y}) > 0$. Many algorithms to solve (3) have been proposed in the literature. The first method was the *fast gradient sign method* (FGSM) (Goodfellow, Shlens, and Szegedy 2015) that takes a single step towards the sign of the adversarial gradient:

$$\mathbf{x}' = \mathbf{x} + \epsilon \ \text{sgn}(\nabla_\mathbf{x} L(\mathbf{x}', y, \hat{y})). \quad (5)$$

Madry et al. (2018) proposed what is commonly referred to as the PGD attack, a method that applies the FGSM step iteratively. This corresponds to running Projected Gradient Descent on the negative adversarial loss:

$$\mathbf{x}^{t+1} = \Pi_{\mathcal{B}(\mathbf{x}, \epsilon)} \left( \mathbf{x}^t + \alpha \ \text{sgn}(\nabla_\mathbf{x} L(\mathbf{x}', y, \hat{y}) \right). \quad (6)$$

To improve the success rate of the attack, we can perform random restarts where we initialize $\mathbf{x}^0 \in \mathcal{B}(\mathbf{x}, \epsilon)$ randomly each time. However, each time we initialize $\mathbf{x}^0$, we ignore all previous unsuccessful attacks as well as the structure of the input domain as well as the optimization problem in general. We propose using an attention mechanism, that significantly reduces the search space thus leading to better initializations and a more efficient attack. Our attention method utilizes Graph Neural Networks (GNNs); we will describe the GNN framework in the next section

## GNN Framework

Our method is motivated by two observations: firstly, neural networks can be interpreted as a graph, with the neurons being nodes, and the weights being edges; secondly common adversarial attacks, such as PGD attack, can be described as taking a forward and a backward pass through this network to generate the adversarial gradient. These observations naturally lead to the idea of using Graph Neural Networks to help with the generation of adversarial examples. The structure of our GNN is based on the neural network we are trying to attack and the message passing algorithm mimics the forward-backward steps used by the PGD attack. Our method is based on work by Lu and Kumar (2020) who use GNNs to make better branching decisions for Neural Network complete verification problems and by Jaeckle and Kumar (2021) who deploy GNNs to generate adversarial examples directly.

## Overview

Given a Neural Network $f$ that we are trying to attack, we create a corresponding Graph Neural Network $G_f = (V, E)$. $V$ is the set of vertices in the GNN; we create one vertex for every neuron in the original network $f$. Similarly,

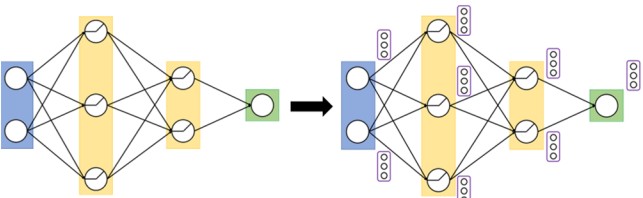

Figure 2: **GNN Framework.** On the left is the original neural network that we're trying to attack, and on the right is the GNN with the embedding vectors initialized for each node. There is a one-to-one correspondence between the nodes of the GNN and the neurons of the original neural network $f$, and between the GNN edges and the NN weights. The nodes are separated into input, hidden, and output layers.

$E$ describes the set of edges, where we have an edge between two nodes if and only if the two corresponding neurons are connected in $f$. We further create a feature vector $\mathbf{z}_v \in \mathbb{R}^p$ for every $v \in V$. These contain important information about the node that we pass to the GNN. Next, we create a multi-dimensional learnt embedding vector $\boldsymbol{\mu}_v$ lying in feature space for every $v \in V$. Once we have created an embedding vector for every node, we start the forward-backward messaging passing algorithm that updates embedding vectors based on information from previous and later layers. Finally, we use another learnt function that takes the embedding vectors corresponding to the nodes in the input layer and outputs a new branching suggestion to decrease our current search space.

## Components

The structure of our GNN including nodes, edges, and node embedding vectors is depicted in Figure 2. We now describe each part in more detail.

**Nodes.** We create one node for every neuron in the original network $f$. We denote the set of nodes in the GNN as $V$.

**Edges.** The set of edges $E$ in the GNN is based on the weights in the original network. The edge weight corresponds to the weights in $f$. The edges influence the message passing algorithm described below.

**Node Features.** We compute a feature vector $\mathbf{z}_v$ for every $v \in V$. We separate all nodes into three categories: input nodes, hidden nodes, and output nodes. We use different methods to generate the node features for each of them. The feature vectors for the input, hidden, and output layers are of dimension $p_{inp}, p_{hid}$, and $p_{out}$ respectively. We aim to use information that encapsulates as much useful information of the respective node as possible, while at the same time being cheap to compute. Some of the features we use include current bounds of the node as well as information from previous unsuccessful PGD attacks. We aim to learn from the unsuccessful attempts to increase the chance of success in the future. This forms a contrasts to the traditional approach of using random restarts that ignore all previous runs. A detailed explanation of how we compute these features can be found in Appendix A.

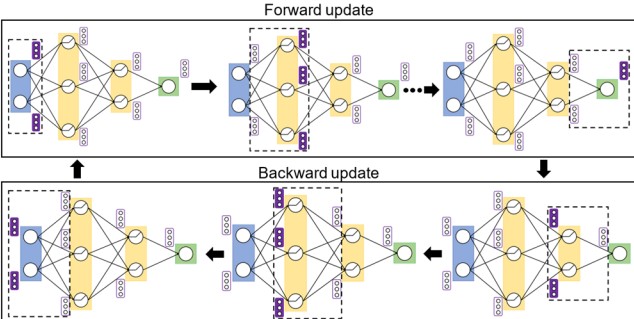

Figure 3: **GNN Message Passing.** The forward update steps are depicted in the top row, and the following backward pass in the bottom row. Embedding vectors are updated using learnt functions that take as input both feature vectors and embedding vectors from previous or later layers respectively. We can perform several rounds of this message passing cycle.

**Node Embeddings.** Embedding vectors are multi-dimensional vectors that lie in feature space, and are generated using learnt functions. We initialize the embedding vectors for the input layer using a learnt function $F_{inp} : \mathbb{R}^p \mapsto \mathbb{R}^d$ that is parameterized by $\boldsymbol{\theta}^0$ and takes as input the feature vector for the input layers ($\mathbf{z}^0$):

$$\boldsymbol{\mu}_j^0 = F_{inp}(\mathbf{z}_j^0; \boldsymbol{\theta}^0). \tag{7}$$

We describe the $F_{inp}$ in greater amount of detail in Appendix B. All other embedding vectors are initialized and updated using functions that take both local feature vectors as well as neighbouring embedding vectors as input. we describe this update procedure in the next subsection. Unlike the feature vectors, the embedding vectors are influenced by neighbouring nodes and thus include information about the state of the entire problem. In particular, once we've completed the message passing algorithm, the embedding vectors of the input layer are influenced by all other embeddings and can thus be used to generate a new branching direction.

### Message Passing

The power of the GNN lies in the message passing algorithm. We initialize and update embedding vectors in a forward-backward manner that is based on the gradient computation procedure used by PGD. We now describe these forward-backward update steps of the GNN in greater detail.

**Forward Pass.** We iteratively update one layer at a time, starting with the first hidden layer. We compute the embedding vector for the $j$-th node in the $i$-th layer using a learnt function $F_{hid} : \mathbb{R}^{p_{hid}+d} \mapsto \mathbb{R}^d$, that takes as input the local feature vectors of the $i$-th layer ($\mathbf{z}_j^i$), the embedding vectors from the previous layer ($\boldsymbol{\mu}^{i-1}$), and the edge matrix ($E$) as follows:

$$\boldsymbol{\mu}_j^i = F_{hid}(\mathbf{z}_j^i, \boldsymbol{\mu}^{i-1}, E; \boldsymbol{\theta}^1) \qquad \forall i \in \{1, \cdots, L-1\}. \tag{8}$$

Next, we compute the embedding vector for the $j$-th node of the output layer, using another learnt function $F_{out}$ :

$\mathbb{R}^{p_{out}+d} \mapsto \mathbb{R}^d$ that takes the local feature vector of the corresponding node ($\mathbf{z}_j^i$) and the embedding vectors from the final hidden layer ($\boldsymbol{\mu}^{L-1}$) to compute an embedding vector for the final layer:

$$\boldsymbol{\mu}_j^L = F_{out}(\mathbf{z}_j^L, \boldsymbol{\mu}^{L-1}; \boldsymbol{\theta}^2). \tag{9}$$

Once we have finished the forward passes, all embedding vectors have been influenced by all embeddings from previous layers, so as long as they corresponding nodes are connected via a path in the original network $f$. As we main focus lies on the embedding vectors of the input layer we now need to send the information backwards.

**Backward Pass.** At this point we have computed an embedding vector for every node in the GNN, and every embedding vector is influenced by embedding vectors from all previous layers. We now perform a backward pass, to send information back from the output layer to the input layer, inspired by the nature of the the back-propagation algorithm that is used to compute the gradient for the PGD attack. For all hidden layers we update the embedding vectors as follows:

$$\boldsymbol{\mu}_j^i = B_{hid}(\mathbf{z}_j^i, \boldsymbol{\mu}^{L+1}; \boldsymbol{\theta}^3) \qquad \forall i \in \{1, \cdots, L-1\}. \tag{10}$$

Like for the forward pass, the backward function takes as input the feature vector, but instead of using the embedding vectors from the previous layer, we now use the embedding vector for the following layer, as we're passing information backwards. Once we have updated all hidden layers we update the embedding vector for the input layer:

$$\boldsymbol{\mu}_j^0 = B_{inp}(\mathbf{z}_j^0, \boldsymbol{\mu}^1; \boldsymbol{\theta}^4). \tag{11}$$

At this point all embedding vectors have been updated based on information from all other layers. We can repeat this forward-backward update scheme if we like. We note that at any point we only need to keep embedding vectors for one single layer in memory. Once we've completed the message passing steps, we need to use the embedding vectors to make a branching decision.

In Appendix B we define and describe the four different functions implementing the forward-backward messaging passing scheme in greater detail. All functions can be implemented efficiently using functions from standard deep learning packages.

### Making a Branching Decision

We now use the embedding vectors of the input layer ($\boldsymbol{\mu}^0$) to output a new branching decision. We aim to split each input node separately. For the $j$-th node, given a lower bound $l_j$ and an upper bound $u_j$ we want to generate new tighter bounds $\bar{l}_j$ and $\bar{u}_j$ such that $l_j \leq \bar{l}_j \leq \bar{u}_j \leq u_j$. We use a learnt function $g : \mathbb{R}^d \mapsto 2$ to get

$$\begin{bmatrix} \hat{l}_j \\ \delta_j \end{bmatrix} = g(\boldsymbol{\mu}_j^0; \boldsymbol{\theta}_5). \tag{12}$$

Here, $\hat{l}_j$ can be interpreted as the new lower bound, and $\delta_j$ as the offset parameter, which defines the difference between

the new upper and lower bounds. To ensure feasibility we take

$$\bar{l}_j \leftarrow \max\{\min\{\bar{l}_j, u_j\}, l_j\}. \qquad (13)$$

This leads to a new lower bound lying in between the old lower and upper bounds. Next we compute the new upper bound $\bar{u}_j$ using both the new lower bound and the offset parameter:

$$\bar{u}_i \leftarrow \max\{\min\{\bar{l}_j + \boldsymbol{\theta}_j, u_i\}, \bar{l}_j\}. \qquad (14)$$

This leads to a new upper bound lying in between the new lower bound and the old upper bound. In the next section we describe how to evaluate the strength of these new bounds and how to train a GNN successfully.

## GNN Training

Having described the GNN framework along with its message passing algorithm, we now show how to train it, to learn to output better branching decisions. We first describe the loss function that we are trying to optimize over and that explains how well our GNN is performing. We then outline the training dataset used to learn the optimal GNN.

### Objective Function

Let us first denote the set of learnable parameters as

$$\boldsymbol{\Theta} = \begin{bmatrix} \boldsymbol{\theta}_0^T & \boldsymbol{\theta}_1^T & \boldsymbol{\theta}_2^T & \boldsymbol{\theta}_3^T & \boldsymbol{\theta}_4^T & \boldsymbol{\theta}_5^T \end{bmatrix}^T. \qquad (15)$$

We use a supervised learning approach to train our GNN. Our training dataset contains adversarial examples that have been generated by successful PGD attacks. We aim to train the GNN to output new bounds that contain these adversarial examples whilst being as tight as possible.

Given an adversarial example $\mathbf{x}_{\text{PGD}}$ returned by a successful PGD attack, and bounds $\bar{l}(\boldsymbol{\Theta})$ and $\bar{u}(\boldsymbol{\Theta})$ outputted by the GNN, we define a loss for each of the $d$ input nodes. The loss consists of two parts. The first one checks whether the adversarial example lies within the bounds:

$$L_{1,i}(\boldsymbol{\Theta}) = \begin{cases} 0 & \text{if } \bar{l}_i(\boldsymbol{\Theta}) \leq x_{\text{PGD},i} \leq \bar{u}_i(\boldsymbol{\Theta}) \\ \bar{l}_i(\boldsymbol{\Theta}) - x_{\text{PGD},i} & \text{if } x_{\text{PGD},i} < \bar{l}_i(\boldsymbol{\Theta}) \\ x_{\text{PGD},i} - \bar{u}_i(\boldsymbol{\Theta}) & \text{if } x_{\text{PGD},i} > \bar{u}_i(\boldsymbol{\Theta}) \end{cases}$$
$$(16)$$

The loss is zero if the adversarial example lies within the bounds and increases linearly with the distance between the true adversarial example and the bounds returned by the GNN. Note, we simplified notation to improve clarity: the output of the GNN $\bar{l}_i(\boldsymbol{\Theta})$ is not only influenced by the learnt parameters $\Theta$, but also by information needed to initialized the feature vectors and the network $f$.

We also want to encourage the bounds to be as tight as possible, in order to significantly reduce the size of the search space. To this end, we define a second loss term that encourages tightness of the bounds:

$$L_{2,i}(\boldsymbol{\Theta}) = \frac{\bar{u}_i(\boldsymbol{\Theta}) - \bar{l}_i(\boldsymbol{\Theta})}{u_i - l_i}. \qquad (17)$$

Our final loss function is a normalized combination of the two sums describing both the tightness of the bounds and whether they include the ground truth:

$$L(\boldsymbol{\Theta}) = \frac{1}{d} \sum_{i=1}^{d} L_{1,i}(\boldsymbol{\Theta}) + \lambda \cdot L_{2,i}(\boldsymbol{\Theta}) \qquad (18)$$

Here, $\lambda$ is a fixed parameter that determines the relative weighting of the two loss functions. If $\lambda$ is small then the GNN focuses on minimizing the first loss term and in the process becomes more conservative to ensure that we don't exclude the ground truth from the new subspace. If, on the other hand, $\lambda$ is large, then the GNN becomes more risky and aims to output a much smaller subspace. We fix the value of $\lambda$ to be 0.033. We will try to minimize this loss using the Adam optimizer (Kingma and Ba 2015) with a learning rate of 1e-4 and no weight decay.

### Training Dataset

We now describe the training dataset we used to learn a successful GNN. It is based on the adversarial training dataset proposed by (Jaeckle and Kumar 2021). They attack a convolutional neural network they call the 'Base' model. It's been trained robustly on the CIFAR10 dataset (Krizhevsky, Hinton et al. 2009) using the methods of Madry et al. (2018) against $l_\infty$ perturbations of size up to $\epsilon = \frac{8}{255}$ (the amount typically considered in empirical works). They created a set of 4515 properties, each a tuple consisting of a natural image ($\mathbf{x}_i$), a true label ($y_i$), an incorrect target label ($\hat{y}_i$), and an allowed perturbation value ($\epsilon_i$). The perturbation value is uniquely chosen for each tuple: it is large enough so that there exist at least one adversarial example in the infinity norm ball around the natural image that it defines; but at the same time small enough so that the adversarial examples are hard to find. The Base network classifies all of these images correctly, so $\epsilon_i > 0$ for all training points $i$.

Before we further describe the training dataset, we remind the reader of the experimental setting that our GNN will be used in: we aim to generate an adversarial example for a given network, and image. If the first PGD run manages to find one, we can move on to the next image. However, if the first iteration of the PGD attack was unsuccessful, we want to use the GNN to learn from this and focus our attention to a smaller input domain, from which we run the next PGD attack. We now need to create a training dataset with which we can simulate this experimental setup. For each data point $(\mathbf{x}_i, y_i, \hat{y}_i,)$ in the training set, we need to generate an adversarial example $\mathbf{x}_{\text{PGD, i}}$ in order to define the GNN objective function (18). We thus need to run PGD repeatedly until we succeed in finding an adversarial example. As mentioned above, the GNN tries to learn from previous unsuccessful attacks by including some of this information in its feature vectors. We there also store information from at least one unsuccessful PGD attack on this data point. Once we have generated a big enough training dataset we can start training a GNN by optimizing over (18).

## Experiments

We now evaluate the performance of our method by comparing it to that of the baseline. We think of our method as a tool

Algorithm 1: PGD Attack with Random Restarts

**Input**: Neural Network $f$, natural image $\mathbf{x}$, true label $y$, incorrect target label $\hat{y}$, perturbation size $\epsilon$
**Parameters**: step size $\alpha$, iteration parameter $T$, restart parameter $R$
**Output**: an adversarial example or *None*

1: **for** $r = 0, \dots, R$ **do**
2:      sample $\mathbf{x}^0$ from $\mathcal{B}(\mathbf{x}, \epsilon)$ uniformly at random
3:      **for** $t = 0, \dots, T$ **do**
4:         **if** $L(\mathbf{x}^t, y, \hat{y}) > 0$ **then**
5:            **Return:** $\mathbf{x}^t$
6:         **else**
7:            $\mathbf{x}^{t+1} = \Pi_{\mathcal{B}(\mathbf{x}, \epsilon)} \left( \mathbf{x}^t + \alpha \, \text{sgn}(\nabla_\mathbf{x} L(\mathbf{x}^t, y, \hat{y})) \right).$
8:         **end if**
9:      **end for**
10:      **Return:** *None*
11: **end for**

that can be applied to existing attacks, rather than create an entirely new one. More specifically, our main aim is to boost the performance of the PGD attack and make it more effective. We thus compare our method against the standard PGD attack using random initializations, to evaluate whether our method is indeed able to improve on the baseline.

**Experimental Set-Up.** We use a similar experimental setup as Jaeckle and Kumar (2021). We run white-box targeted image-dependent adversarial attacks on the CIFAR10 dataset (Krizhevsky, Hinton et al. 2009). We attack a convolutional neural network they call the 'Base' model: it consists of two convolutional layers followed by two fully connected ones and use the ReLU activation function. We attack 100 properties, all different to the ones seen during training. We report the percentage of properties successfully attacked over time, both for the baseline and our method. We use this particular dataset, as it only consists of challenging adversarial properties. As is the case for the training dataset, every data point in the test set is a tuple consisting of a natural image $(\mathbf{x}_i)$, a true label $(y_i)$, an incorrect target label $(\hat{y}_i)$, and an allowed perturbation value $(\epsilon_i)$. The perturbation norm is image dependent to ensure that at least one adversarial example exists in $\mathcal{B}(\mathbf{x}, \epsilon)$, it's small enough so that it is difficult to find. Both the baseline and our method are implemented in Pytorch (Paszke et al. 2017) and are run on a single GPU each.

**Baseline.** We compare our method against the standard PGD attack (Madry et al. 2018) with random initializations. PGD tries to find an adversarial example by first choosing a starting point $\mathbf{x}^0 \in \mathcal{B}(\mathbf{x}, \epsilon)$ uniformly at random and then running the following update step for 100 iterations:

$$\mathbf{x}^{t+1} = \Pi_{\mathcal{B}(\mathbf{x}, \epsilon)} \left( \mathbf{x}^t + \alpha \, \text{sgn}(\nabla_\mathbf{x} L(\mathbf{x}', y, \hat{y})) \right). \quad (19)$$

We stop early if we've found an adversarial example, that is if for any $t$, we get $L(\mathbf{x}', y, \hat{y}) > 0$. We pick $\alpha = 0.1$ and perform a total of 210 restarts or until we've found an adversarial example. The baseline method is further described in Algorithm 1.

Algorithm 2: PGD Attack with GNN Attention

**Input**: Neural Network $f$, natural image $\mathbf{x}$, true label $y$, incorrect target label $\hat{y}$, perturbation size $\epsilon$
**Parameters**: step size $\alpha$, iteration parameter $T$, restart parameter $R$, GNN parameters $\boldsymbol{\Theta}$
**Output**: an adversarial example or *None*

1: initialize an empty dictionary dict to store information of the PGD attack for the GNN
2: sample $\mathbf{x}^0$ from $\mathcal{B}(\mathbf{x}, \epsilon)$ uniformly at random
3: **for** $t = 0, \dots, T$ **do**
4:      **if** $L(\mathbf{x}^t, y, \hat{y}) > 0$ **then**
5:         **Return:** $\mathbf{x}^t$
6:      **else**
7:         $\mathbf{x}^{t+1} = \Pi_{\mathcal{B}(\mathbf{x}, \epsilon)} \left( \mathbf{x}^t + \alpha \, \text{sgn}(\nabla_\mathbf{x} L(\mathbf{x}^t, y, \hat{y})) \right).$
8:         add $L(\mathbf{x}^t, y, \hat{y})$ and $\mathbf{x}^{t+1}$ dict
9:      **end if**
10: **end for**
11: Initialize feature vectors $\mathbf{z}$ using $(dict, f, \mathbf{x}, y, \hat{y}, \mathcal{B}(\mathbf{x}, \epsilon))$ as described in Appendix A
12: Initialize and update embedding vectors $\boldsymbol{\mu}$ using the forward-backward message passing algorithm defined by equations (7) - (11) and in Appendix B
13: Generate a new bounded input set $B_{GNN}$ using equations (12) - (14)
14: **for** $r = 0, \dots, R$ **do**
15:      sample $\mathbf{x}^0$ from $B_{GNN}$ uniformly at random
16:      **for** $t = 0, \dots, T$ **do**
17:         **if** $L(\mathbf{x}^t, y, \hat{y}) > 0$ **then**
18:            **Return:** $\mathbf{x}^t$
19:         **else**
20:            $\mathbf{x}^{t+1} = \Pi_{B_{GNN}} \left( \mathbf{x}^t + \alpha \, \text{sgn}(\nabla_\mathbf{x} L(\mathbf{x}^t, y, \hat{y})) \right).$
21:         **end if**
22:      **end for**
23:      **Return:** *None*
24: **end for**

**Our Method.** Our method consists of three parts. We first run PGD with no restarts with the same hyper-parameters as above. If the attack has been successful we move on to the next image. If unsuccessful we then use the GNN, that has been trained as described in the previous section. The GNN takes as input the current bounds, the image we are trying to attack and data from the unsuccessful PGD attack, to output new, tighter bounds. We then run PGD on the new bounds a further 99 times or until we have found an adversarial example. We perform two iterations of the forward-backward message passing procedure described above as the embedding vectors tend to converge after two passes. In Algorithm 2 we summarize the entire algorithm, including the initial PGD attack, the execution of the GNN, and the final PGD attacks with the new GNN computed bounds. In the Appendix, we explore different variations of our method, including calling our method up to 6 times after a varying number of restarts to further decrease the search space and learn from more unsuccessful attacks. We also describe the hyper-parameter used for the GNN.

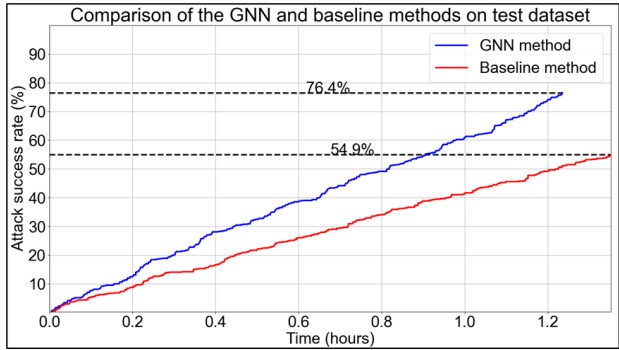

Figure 4: Comparing our method against PGD with random restarts. We note that for any given time, our method outperforms the baseline, finding adversarial examples for 40% more properties.

**Results.** We compare our method against the baseline in Figure 4. Our method significantly boost the performance PGD compared to when using random initializations. We increase the number of properties successfully attacked by over 35% while decreasing the average time taken to do so. One downside of our method compared to using random initializations is the one-off cost associated with training the GNN. However, we argue that in most applications, such as verification or adversarial training, we don't just call the adversarial attack once, but a large number of times. The improved performance of the GNN thus makes up for the one time training cost.

## Conclusion

In this work we have shown how to improve an existing adversarial attack method using a Graph Neural Network as an attention mechanism that learns from previous unsuccessful PGD attacks and greatly decreases the search space for future attacks. By improving the starting point for the next attack, PGD has a higher chance of converging to find an adversarial example in less time. Our method leads to a 39% increase in the attack success rate for a given timeout, compared to using PGD with random initializations.

Being able to compute adversarial examples efficiently is important for making neural networks more robust and improving explainability; both are important for their application in safety-critical situations and form an important area of research.

There is a lot of potential for future work to build on our method and to extend it. This could include using a similar approach to work for other attack methods that currently use random initializations, such as the the Carlini Wagner attack (Carlini and Wagner 2017), MI-FGSM, the iterative fast gradient sign method with momentum (Dong et al. 2018), or autoattack (Croce and Hein 2020). One could also combine it with other attack methods that use learning to output adversarial examples such as AdvGAN (Poursaeed et al. 2018), ATN (Baluja and Fischer 2017), or AdvGNN (Jaeckle and Kumar 2021). Moreover, the GNN based approach could be extended to work on larger or deeper neural networks, or

those containing residual connections.

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

# APPENDIX

We now present the appendix supplementing the main paper. In section A we describe in greater detail how the node feature vectors are created and in section B we show how the message passing functions are implemented.

## A Node features

This section explains how the feature vectors can be constructed for the input, hidden and output layers. As mentioned above, we aim to design the features so that they capture the maximum possible information needed to make a good branching decision while keeping the computational complexity as low as possible.

### Input node features

Firstly, the original lower and upper bounds on the $i$-th input node $l_i$ and $u_i$ are selected as this node's features as they are very indicative of the influence of this node on the output of the network.

If the GNN framework is invoked after some initial adversarial PGD attack is unsuccessful, then the information obtained from this attack can be used to generate more features for the nodes of the network. It should be noted that if this initial adversarial PGD attack is successful, then the GNN framework does not need to be invoked at all since the given property was proven to be false by a single randomly initialised PGD attack. Otherwise, all the available information can be provided to the GNN for it to make a branching decision. The first and obvious choice of a feature which can easily be obtained for each input node from an unsuccessful PGD attack is the value of this node at the end of the PGD attack.

Finally, information about the gradients of the output of a given neural network with respect to all of its inputs throughout the unsuccessful adversarial PGD attack can also be indicative of the location of a valid adversarial attack within the input domain. One option would be to provide the gradients at each step of the unsuccessful PGD attack to the GNN. However, since the number of steps of a PGD attack can vary and sometimes be in the order of thousands or tens of thousands, it might be excessive to do that as this will introduce redundancy and lead to higher computational complexity which is undesirable. Hence, in this project the following information about the gradients of the network's output with respect to each input was selected to enter the feature vectors of the corresponding input nodes:

- The mean gradient over all steps of an unsuccessful PGD attack

- The median gradient over all steps of an unsuccessful PGD attack

- The maximum gradient over all steps of an unsuccessful PGD attack

- The minimum gradient over all steps of an unsuccessful PGD attack

- The standard deviation of the gradients over all steps of an unsuccessful PGD attack

- The gradient at the last step of an unsuccessful PGD attack

## Hidden node features

The features of all the activation nodes of a given network should be designed to contain information about the propagation of the numerical values through the network which in its turn can help the GNN to deduce the overall effect of each of the network's inputs on its output. As in the case of the input node features, it is sensible to include the lower and upper bounds on each activation node in its feature vector. However, while the exact lower and upper bounds on the input nodes are directly available from input constraints, obtaining the bounds on all the activation nodes is much more difficult. Computing the exact minima and maxima of the nodes of the activation and output layers is in general an intractable problem which means that approximate solutions have to be used to calculate the reasonably tight lower and upper bounds on these nodes. We compute these intermediate bounds using linear bound relaxations (Weng et al. 2018) This function made it possible to obtain the approximate lower and upper bounds for all the activation and output layer nodes given the bounds on all the input layer nodes.

Since the unsuccessful PGD attack after which the GNN framework is initialised contains the values for each node of each layer at the end of the attack, a third feature which should enter the feature vectors of all the activation nodes is the value of the corresponding node at the end of the PGD attack.

The fourth feature which describes each activation node quite well and hence should be included in its feature vector is its associated bias in the original network. Mathematically, for $i$-th node of $k$-th activation layer where $k \in \{1, 2, ..., L-1\}$, the bias which should enter its feature vector is given by $b_i^k$.

The fifth feature is based on relaxations of the ReLU nonlinearity and is taken from work by (Lu and Kumar 2020). Activation nodes are quite specific due to their nonlinear nature and so specifying all of the above features might still not be enough to fully describe them. In case of the ReLU activation function, the state of any activation node can belong to one of the three cases. Denoting the lower and upper bounds of the $i$-th node of the $k$-th activation layer as as $l_i^k$ and $u_i^k$ respectively and the node values before and after the ReLU function is applied as $\widehat{x}_i^k$ and $x_i^k$ respectively, these cases can be visualised as shown in Figure 5. In all parts of the figure, the ReLU function is plotted in blue and its part being considered based on the values of the lower and upper bounds is indicated in red.

In the first case, shown on the left, both lower and upper bounds on a particular activation node happen to be non-positive. Such activation node is referred to be in its blocking state as it has zero as its output for all possible inputs given by the lower and upper bounds, i.e. it blocks all the information. In the same way as it was done in (Lu and Kumar 2020), the final state which measures ambiguity, denoted by $\beta_i^k$, of each such activation node will be given by $\beta_i^k = 0$ as there is no ambiguity associated with this node.

The second case, shown in the middle, arises when both lower and upper bounds on a particular activation node are

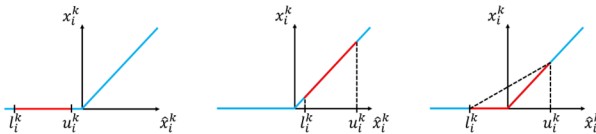

Figure 5: Three cases of the state of a ReLU activation node based on this node's bounds. Left: blocking state due to both lower and upper bounds on a node being non-positive in which case the output of such ReLU activation node is always zero. Middle: passing state due to both lower and upper bounds on a node being non-negative in which case the output of such node is always equal to its input. Right: ambiguous state due to the lower and upper bounds being negative and positive respectively in which case the output can be either zero or equal to the input depending on the exact input to the node.

non-negative. In this case, the activation node is referred to be in its passing state because this time its output will be equal to its input for all possible inputs given by the lower and upper bounds. The final state of such activation node is also given by $\beta_i^k = 0$ because, as in the first case, there is no ambiguity involved in the propagation of information through such activation node.

In the final case, shown on the right, the lower and upper bounds on a particular activation node turn out to be negative and positive respectively. The state of such activation node is referred to as ambiguous since its output can be either zero or equal to its input depending on the exact input value. To define $\beta_i^k$, the same approach which was used in (Lu and Kumar 2020) can be followed whereby the intercept of the triangle formed by the section of the ReLU between $l_i^k$ and $u_i^k$ and the line connecting the ends of this section, indicated by the dashed line in the figure, is considered. The equation of this line can easily be shown to be:

$$x_i^k(\widehat{x}_i^k) = \frac{u_i^k}{u_i^k - l_i^k}\widehat{x}_i^k - \frac{u_i^k l_i^k}{u_i^k - l_i^k} \tag{20}$$

As $u_i^k \to 0$ and/or $l_i^k \to 0$, the intercept of the above line tends to zero, and as $u_i^k$ becomes more positive and/or $l_i^k$ becomes more negative, it increases. Hence, the intercept of the above line is a suitable measure of ambiguity of a ReLU node and therefore for each ambiguous activation node:

$$\beta_i^k = -\frac{u_i^k l_i^k}{u_i^k - l_i^k} \tag{21}$$

where $\beta_i^k$ in the equation above is necessarily positive since for an ambiguous node $u_i^k > 0$ and $l_i^k < 0$.

## Output node features

The output node features need to be informative of the state of the output of a given neural network to provide the GNN with the information about the input-output relationship of the network. The output feature vectors can be constructed using the same features as in the case of the activation nodes apart from the ambiguity descriptor. Hence, four features to enter the feature vector of each output node are:

- Lower bound on the output node
- Upper bound on the output node
- Node value at the end of an unsuccessful PGD attack
- Node bias in the original network

Having described in greater detail how we compute the node feature vectors $\mathbf{z}$, we now turn our attention to the implementation of the message passing algorithm.

## B  Message Passing Functions

There are 5 learnt functions that implement the GNN message passing algorithm: $F_{inp}$, $F_{hid}$, $F_{out}$, $B_{hid}$, $B_{inp}$; and one learnt function that turns the embedding vectors into a branching decision: $g$. Some of these functions are inspired by the work of Lu and Kumar (2020). Each of the six functions will now be discussed in detail in turn. From here on we refer to $F_{hid}$ and $B_{hid}$ as $F_{hid}$ and $B_{hid}$ respectively, to highlight that the state of the activation function, in our case the

### Forward pass — input layer

The first network of interest is the one corresponding to the function $F_{inp}$. It has already been mentioned in the Section that it, along with all the other networks, has a form of a multi-layered fully-connected network. This network in particular should simply process a local feature vector of the given input node and return an updated input embedding vector. Hence, this neural network will be designed to have a single stage with one activation layer containing ReLU activation functions, as depicted in Figure 6.



Figure 6: Structure of the forward input embedding vector update network

It is important to make a few assumptions for the simplicity of the design of all the auxiliary neural networks, including the one above, in accordance with (Lu and Kumar 2020). Firstly, it will be assumed that all the embedding vectors, i.e. the ones of the input, activation and output nodes, are of the same size. Secondly, all the activation layers of all the auxiliary neural networks will be assumed to have the same size. Finally, the two sizes from the previous two points will be assumed to be equal, as shown in Figure 6.

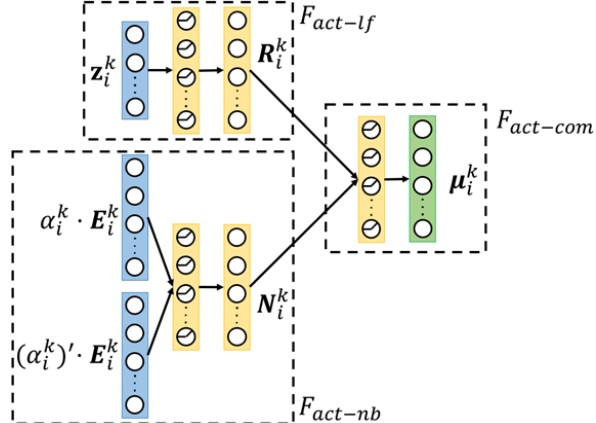

Figure 7: Structure of the forward activation embedding vector update network

### Forward pass — hidden layer

The network corresponding to the function $F_{act}$ should take a local feature vector of the activation node as well as all the embedding vectors of all the previous layer nodes propagated to this node as inputs and return an updated activation embedding vector. In contrast to the simple network described in Subsection B, in this case there are three stages to be considered which are explained below in turn.

The overall structure of the neural network which implements the update function $F_{act}$ is shown in Figure 7 where, since the network is fully-connected, the connections between layers are illustrated by single arrows for better visualisation purposes. The first stage, appearing in the top left corner of the figure and denoted by $F_{act-lf}$, is meant to process the local features of the $i$-th node of the $k$-th activation layer where $k \in \{1, 2, ..., L-1\}$. This stage has the exact same structure as the network implementing $F_{inp}$. It should be noted at this point that a further assumption has to be made for simplicity which says that the outputs of all the intermediate stages of all the auxiliary neural networks should have their size equal to that of the activation layers of these networks, as shown in Figure 7. Denoting the parameters of the first stage of the network as $\boldsymbol{\theta}_1^0$ and its output as $\boldsymbol{R}_i^k$, the operation of $F_{act-lf}$ can be written as:

$$\boldsymbol{R}_i^k = \begin{cases} F_{act-lf}(\mathbf{z}_i^k; \boldsymbol{\theta}_1^0) & \text{if} \quad \beta_i^k > 0 \\ \mathbf{0} & \text{otherwise} \end{cases} \tag{22}$$

where it is important to note that the condition for the first statement means that the pass through the first stage should only be made if the node is ambiguous, as explained in Subsection A. Otherwise, the output from the first stage is set to the zero vector of appropriate size, in accordance with (Lu and Kumar 2020).

The second stage of the network, denoted by $F_{act-nb}$ and appearing in the bottom left corner of Figure 7, needs to process the embedding vectors of the previous layer, i.e. those of the neighbouring nodes, hence the subscript. To do that, these should first be propagated forward and combined at the current node of the current activation layer.

Considering the $i$-th node of the $k$-th activation layer where $k \in \{1, 2, ..., L-1\}$ and the weight matrix $\mathbf{W}_k$ connecting this layer with the previous $(k-1)$-th layer, this is done by taking $j$-th embedding vector of the previous layer in turn, multiplying it by the corresponding weight $W_{i,j}^k$ and then summing over $j$. Mathematically, the resulting vector which contains information about the combined embedding vectors of the previous layer, denoted by $\boldsymbol{E}_i^k$, can be computed as:

$$\boldsymbol{E}_i^k = \sum_j W_{i,j}^k \cdot \boldsymbol{\mu}_j^{k-1} \qquad (23)$$

where it is very important to note that, in contrast to the conventional pass through the original neural network, the bias of the $i$-th node of the $k$-th activation layer is not applied when computing $\boldsymbol{E}_i^k$.

Once $\boldsymbol{E}_i^k$ is computed, the final processing step involves considering the amount of information which passes through the activation node. By looking at the equation (20) from Subsection A, it can be seen that as $u_i^k \to 0$, i.e. as the node tends to its blocking state, the slope of this equation, given by $\frac{u_i^k}{u_i^k - l_i^k}$, tends to 0. On the other hand, as $l_i^k \to 0$, i.e. as the node tends to its passing state, the slope tends to 1. When the node is in its ambiguous state, however, the slope lies in the range $(0, 1)$. Hence, the slope of (20), which will be denoted by $\alpha_i^k$, is a suitable and well-defined measure of information passing through the node. Using the method from (Lu and Kumar 2020) and denoting the parameters of the second stage network as $\boldsymbol{\theta}_1^1$ and its output as $\boldsymbol{N}_i^k$, the operation of $F_{act-nb}$ can be mathematically defined as follows:

$$\boldsymbol{N}_i^k = F_{act-nb}\left( \begin{bmatrix} \alpha_i^k \cdot (\boldsymbol{E}_i^k)^T & (\alpha_i^k)' \cdot (\boldsymbol{E}_i^k)^T \end{bmatrix}^T ; \boldsymbol{\theta}_1^1 \right) \qquad (24)$$

where the two vectors are concatenated to form one vector of double the size and $(\alpha_i^k)'$ is defined as:

$$(\alpha_i^k)' = \begin{cases} 1 - \alpha_i^k & \text{if} \quad 0 < \alpha_i^k < 1 \\ \alpha_i^k & \text{otherwise} \end{cases} \qquad (25)$$

The third and final stage of the network, denoted by $F_{act-com}$ and appearing on the right in Figure 7, combines the information obtained from the local features, given by $\boldsymbol{R}_i^k$, and that from the neighbouring embedding vectors, given by $\boldsymbol{N}_i^k$, to return the updated embedding vector of a particular activation node. Denoting the parameters of this stage as $\boldsymbol{\theta}_1^2$, its operation can be defined in the following way for all $k \in \{1, 2, ..., L-1\}$:

$$\boldsymbol{\mu}_i^k = F_{act-com}\left( \begin{bmatrix} (\boldsymbol{R}_i^k)^T & (\boldsymbol{N}_i^k)^T \end{bmatrix}^T ; \boldsymbol{\theta}_1^2 \right) \qquad (26)$$

**Forward pass — output layer**

The final forward update network is the one associated with the function $F_{out}$. Its operation is completely analogous to that of $F_{act}$ since it has to take a local output feature vector as well as all the embedding vectors of the last activation layer nodes propagated forward as inputs and return an

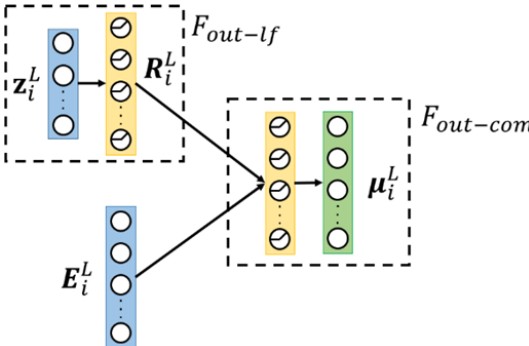

Figure 8: Structure of the forward output embedding vector update network

updated output embedding vector. This is achieved by a network structure involving two stages, as shown in Figure 8. The first stage, denoted by $F_{out-lf}$ and appearing in the top left corner of the figure, processes the local output features and has almost the same structure as the network implementing $F_{act-lf}$ from Figure 7. Denoting the parameters of the first stage as $\boldsymbol{\theta}_2^0$ and its output as $\boldsymbol{R}_i^L$, its operation is defined as:

$$\boldsymbol{R}_i^L = F_{out-lf}(\mathbf{z}_i^L; \boldsymbol{\theta}_2^0) \qquad (27)$$

Since the ReLU activation function is not applied to the output nodes of the original neural network, there is no need for either a conditional statement in the equation above or for further processing applied to the embedding vectors of the last activation layer once they are propagated forward and combined to form the vector $\boldsymbol{E}_i^L$. This vector is obtained in exactly the same way as before according to equation (23). Hence, the second and last stage of this update network, denoted by $F_{out-com}$ and shown on the right in Figure 8, concatenates $\boldsymbol{R}_i^L$ directly with $\boldsymbol{E}_i^L$ to produce an updated embedding vector at its output. Denoting the parameters of this stage as $\boldsymbol{\theta}_2^1$, its operation can be defined in the following way:

$$\boldsymbol{\mu}_i^L = F_{out-com}\left( \begin{bmatrix} (\boldsymbol{R}_i^L)^T & (\boldsymbol{E}_i^L)^T \end{bmatrix}^T ; \boldsymbol{\theta}_2^1 \right) \qquad (28)$$

**Backward pass — hidden layer**

The first neural network which performs the backward update on the embedding vectors is the one corresponding to the update function $B_{act}$. The operation of this function is very similar to that of $F_{act}$ in that it also takes a local feature vector of a particular activation node as one of the inputs and returns an updated embedding vector for this node. The second input, however, unlike in the case of $F_{act}$, should consist of the propagated and combined embedding vectors of the next rather than previous layer nodes. In addition, due to the features selected in Subsection A for the activation nodes in this project being slightly different to the features selected in (Lu and Kumar 2020), the design of the network implementing $B_{act}$ will also be a bit different. This network,

similarly to $F_{act}$, has three stages to it which are all shown in Figure 9.

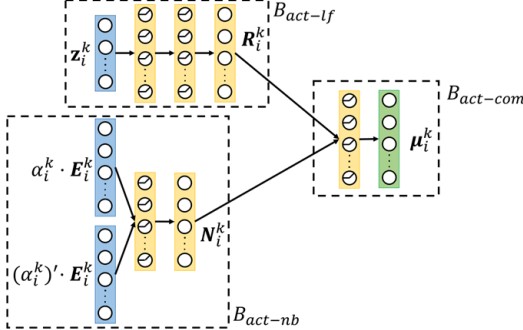

Figure 9: Structure of the backward activation embedding vector update network

The first stage of this network, denoted by $B_{act-lf}$, is completely analogous to $F_{act-lf}$. Denoting the parameters of $B_{act-lf}$ as $\boldsymbol{\theta}_3^0$ and its output, similarly to the case of the first stages of the forward update networks, as $\boldsymbol{R}_i^k$ where $k \in \{1, 2, ..., L-1\}$, the operation of the first stage can be defined as follows:

$$\boldsymbol{R}_i^k = \begin{cases} B_{act-lf}(\mathbf{z}_i^k; \boldsymbol{\theta}_3^0) & \text{if } \beta_i^k > 0 \\ \mathbf{0} & \text{otherwise} \end{cases} \quad (29)$$

where the condition for the first statement is the same as in case of equation (22) and implies that the pass through the network defined by $B_{act-lf}$ is only made if the activation node under consideration is ambiguous whereas otherwise the output from this stage is set to the zero vector of the appropriate size.

The structure of the second stage, denoted by $B_{act-nb}$, is exactly the same as the one of $F_{act-nb}$. The only difference is that to form the vector $\boldsymbol{E}_i^k$, the embedding vectors of all the nodes of the next layer now have to be propagated backwards and combined. Using the same notation as in Subsection B and again noting that bias is not applied when propagating embedding vectors, $\boldsymbol{E}_i^k$ is obtained as:

$$\boldsymbol{E}_i^k = \sum_j W_{i,j}^{k+1} \cdot \boldsymbol{\mu}_i^{k+1} \quad (30)$$

Denoting the parameters of the second stage of the network as $\boldsymbol{\theta}_3^1$ and its output, similarly to the case of the output of $F_{act-nb}$, as $\boldsymbol{N}_i^k$, its operation is then defined in the same way as the operation of $F_{act-nb}$:

$$\boldsymbol{N}_i^k = B_{act-nb}\left( \begin{bmatrix} \alpha_i^k \cdot (\boldsymbol{E}_i^k)^T & (\alpha_i^k)' \cdot (\boldsymbol{E}_i^k)^T \end{bmatrix}^T ; \boldsymbol{\theta}_3^1 \right) \quad (31)$$

where $\alpha_i^k$ and $(\alpha_i^k)'$ are defined in the same way as before.

The third and final stage of this network, denoted by $B_{act-com}$, is again the same as the one associated with $F_{act-com}$. Denoting its parameters as $\boldsymbol{\theta}_3^2$, $B_{act-com}$ is defined can be the following way or $k \in \{1, 2, ..., L-1\}$:

$$\boldsymbol{\mu}_i^k = B_{act-com}\left( \begin{bmatrix} (\boldsymbol{R}_i^k)^T & (\boldsymbol{N}_i^k)^T \end{bmatrix}^T ; \boldsymbol{\theta}_3^2 \right) \quad (32)$$

## Backward pass — input layer

The second network which performs the backward update on the embedding vectors and concludes one round of updates is the one associated with the function $B_{inp}$. In the same way the network implementing $F_{out}$ was similar to the one implementing $F_{act}$, this network is very similar to the previous one implementing $B_{act}$. It should take a local feature vector of an input node together with all the embedding vectors of the first activation layer nodes propagated backwards as before and return an updated embedding vector for this input node. The structure of the network implementing $B_{inp}$ consists of two stages, as shown in Figure 10, and can be observed to be almost identical to the one which appeared in Figure 8.

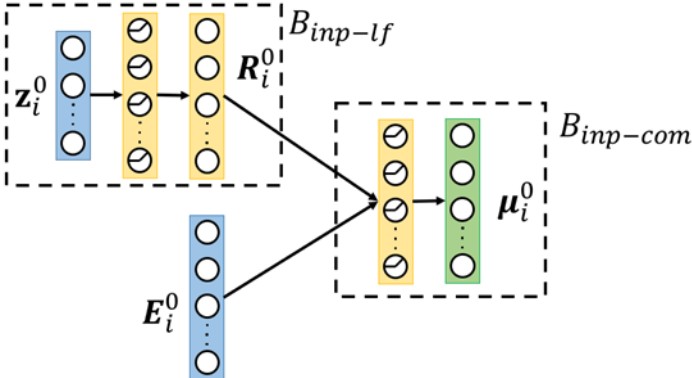

Figure 10: Structure of the backward input embedding vector update network

The first stage of the above neural network, denoted by $B_{inp-lf}$, processes the local feature vector of the given input node in a similar way to all the networks mentioned above. Denoting the parameters of this stage as $\boldsymbol{\theta}_4^0$ and its output as $\boldsymbol{R}_i^0$, the operation of $B_{inp-lf}$ can be defined in the following way:

$$\boldsymbol{R}_i^0 = B_{inp-lf}(\mathbf{z}_i^0; \boldsymbol{\theta}_4^0) \quad (33)$$

Again, since the ReLU activation functions are not applied at the input nodes, there is no need for either a conditional statement in the equation above or for further processing of the vector $\boldsymbol{E}_i^0$ of the propagated backwards and combined embedding vectors of the first activation layer nodes. The vector $\boldsymbol{E}_i^0$ is obtained in the same way as before in accordance with equation (30). The second stage of $B_{inp}$ then becomes identical to the second stage of $F_{out}$ so the vectors $\boldsymbol{R}_i^0$ and $\boldsymbol{E}_i^0$ can be directly concatenated and passed through this stage, denoted by $B_{inp-com}$, to obtain the updated embedding vector of a particular input node. Denoting the parameters of this stage as $\boldsymbol{\theta}_4^1$, $B_{inp-com}$ is defined as:

$$\boldsymbol{\mu}_i^0 = B_{inp-com}\left( \left[ (\boldsymbol{R}_i^0)^T \quad (\boldsymbol{E}_i^0)^T \right]^T ; \boldsymbol{\theta}_4^1 \right) \qquad (34)$$

**Branching decision**

The last auxiliary neural network involved in the GNN framework is the one which implements the function $g$ by computing the new lower bound $\bar{l}_i$ and $\bar{\delta}_i$, the offset from it, which defines the new upper bound for each input node. By doing so, $g$ can potentially greatly reduce the search space where a valid PGD attack is most likely to be and thus make the future PGD attacks more likely to succeed. The network implementing $g$, the structure of which is shown in Figure 11, has only one stage which takes an embedding vector of a particular input node and returns a two-dimensional vector of $\bar{l}_i$ and $\bar{\delta}_i$.

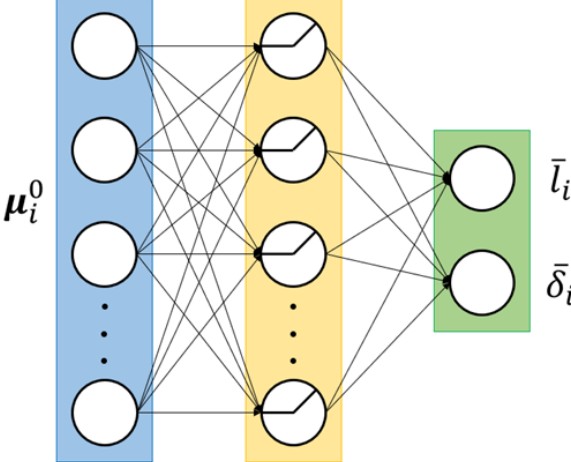

Figure 11: Structure of the bounds update network

