# OpenReview forum: "Attention for Adversarial Attacks: Learning from your Mistakes"
_AAAI.org/2022/Workshop/AdvML — AAAI-22 AdvML Workshop Oral_

### Official Review · Reviewer_aczm · 2021-11-27
**A good idea to introduce GNN into adversarial attacks. The efficiency of regular adversarial attackers is improved.**

**Rating:** 8
**Confidence:** 4

**Review:**

Pros:
1. The writing is good.
2. The explanation of using GNN is clear.
3. The experimental results are convincing.
4. Using GNN as an attention mechanism to find the pitfalls of a classifier is a good idea. Maybe it can be used in more important scenarios.

---

### Official Review · Reviewer_6hzb · 2021-11-29
**Review for adversarial attacks with attention**

**Rating:** 7
**Confidence:** 4

**Review:**

This paper introduces a white box adversarial attack method using attention to refine the PGD attack. A GNN is trained on adversarial examples generated by PGD to narrow down the search space of PGD. The GNN is used in the attack algorithm to learn from possible 'bad' start points. Experiments show that the proposed method increase the attack efficiency and success rate compared to the baseline method.

---

### Decision · Program_Chairs · 2021-12-01

**Decision:**

Accept (Oral)

**Comment:**

Both reviewers have high ratings on this paper. Thus it is accepted with an oral presentation.